# Differential Drought Responses of Soybean Genotypes in Relation to Photosynthesis and Growth-Yield Attributes

**DOI:** 10.3390/plants13192765

**Published:** 2024-10-02

**Authors:** Md. Saddam Hossain, Md. Arifur Rahman Khan, Apple Mahmud, Uttam Kumar Ghosh, Touhidur Rahman Anik, Daniel Mayer, Ashim Kumar Das, Mohammad Golam Mostofa

**Affiliations:** 1Department of Agronomy, Bangabandhu Sheikh Mujibur Rahman Agricultural University, Gazipur 1706, Bangladesh; saddam.agr@bsmrau.edu.bd (M.S.H.); apple885@bsmrau.edu.bd (A.M.); uttam@bsmrau.edu.bd (U.K.G.); 2Department of Plant and Soil Science, Texas Tech University, Lubbock, TX 79409, USA; tanik@ttu.edu (T.R.A.); daniel.mayer@ttu.edu (D.M.); 3Department of Applied Biosciences, College of Agriculture and Life Science, Kyungpook National University, Daegu 41566, Republic of Korea; ashim@knu.ac.kr; 4Department of Chemistry, State University of New York College of Environmental Science and Forestry, Syracuse, NY 13210, USA

**Keywords:** drought, soybean genotypes, gas exchange features, proline, stress tolerance

## Abstract

Water scarcity leads to significant ecological challenges for global farming production. Sustainable agriculture depends on developing strategies to overcome the impacts of drought on important crops, including soybean. In this present study, seven promising soybean genotypes were evaluated for their drought tolerance potential by exposing them to water deficit conditions. The control group was maintained at 100% field capacity (FC), while the drought-treated group was maintained at 50% FC on a volume/weight basis. This treatment was applied at the second trifoliate leaf stage and continued until maturity. Our results demonstrated that water shortage exerted negative impacts on soybean phenotypic traits, physiological and biochemical mechanisms, and yield output in comparison with normal conditions. Our results showed that genotype G00001 exhibited the highest leaf area plant^−1^ (483.70 cm^2^), photosynthetic attributes like stomatal conductance (*gs*) (0.15 mol H_2_O m^−2^ s^−1^) and photosynthetic rate (*Pn*) (13.73 μmol CO_2_ m^−2^ s^−1^), and xylem exudation rate (0.25 g h^−1^) under drought conditions. The G00001 genotype showed greater leaf greenness by preserving photosynthetic pigments (total chlorophylls (Chls) and carotenoids; 4.23 and 7.34 mg g^−1^ FW, respectively) in response to drought conditions. Soybean plants accumulated high levels of stress indicators like proline and malondialdehyde when subjected to drought stress. However, genotype G00001 displayed lower levels of proline (4.49 μg g^−1^ FW) and malondialdehyde (3.70 μmol g^−1^ FW), indicating that this genotype suffered from less oxidative stress induced by drought stress compared to the other investigated soybean genotypes. Eventually, the G00001 genotype had a greater yield in terms of seeds pod^−1^ (SP) (1.90) and 100-seed weight (HSW) (14.60 g) under drought conditions. On the other hand, BD2333 exhibited the largest decrease in plant height (37.10%), pod number plant^−1^ (85.90%), SP (56.20%), HSW (54.20%), *gs* (90.50%), *Pn* (71.00%), transpiration rate (59.40%), relative water content (34.40%), Chl *a* (79.50%), total Chls (72.70%), and carotenoids (56.70%), along with the maximum increase in water saturation deficit (290.40%) and malondialdehyde content (280.30%) under drought compared to control conditions, indicating its higher sensitivity to drought stress. Our findings suggest that G00001 is a promising candidate to consider for field trials and further evaluation of its molecular signature may help breeding other elite cultivars to develop drought-tolerant, high-yielding soybean varieties.

## 1. Introduction

Drought, characterized by water scarcity in the plant rhizosphere, stands as the foremost abiotic stress severely impacting global food production [1]. This challenge is exacerbated by increasing population growth, unpredictable rainfall patterns, and rising temperatures linked to climate change. Collectively, these factors intensify water scarcity conditions in agriculture [2]. The repercussions of water scarcity on crops are profound, changing their phenotypic traits, physiological and biochemical processes, and ultimately crop yield [1,3,4,5,6]. Drought negatively impacts plant morphology such as reduced leaf number and leaf growth, as well as decreased shoot height and seed quantity [1]. Physiological effects of drought stress include reduced cell turgor pressure, which inhibits cell expansion and plant growth [7]. Moreover, drought conditions lead to diminished leaf gas exchange and a decreased leaf area index, further limiting biomass accumulation in both roots and shoots [8,9]. These effects are exacerbated if soybean plants are exposed to water shortage during reproductive stages, as this can directly affect flowering and pod formation [10,11]. In response to water scarcity, plants have evolved adaptive strategies such as the deepening of root systems to access deeper soil water reserves, reduced transpiration rates to minimize water loss, and improved water use efficiency [12,13,14]. Additionally, plants accumulate compatible organic compounds like proline, sugars, polyols, and amino acids to sustain osmotic balance under water shortage conditions [15,16,17,18,19]. Plants also deploy complex antioxidant defense systems to mitigate oxidative stress induced by over accumulation of reactive oxygen species (ROS) under water-deprived conditions [15,20,21].

Soybean (*Glycine max*) is a major oilseed crop cultivated across the world. Soybean has a great nutritional value (40–45% protein, 18–21% oil, and 26–30% carbohydrates) [17,22]. The crude oil extracted from soybeans is made up of roughly 55% linoleic acid, 21% oleic acid, 12% palmitic acid, 9% linolenic acid, and 4% stearic acid [23]. Globally, about 70% of plant proteins and 29% of edible oil is extracted from soybean [17]. It has also an economic significance as it provides raw materials for animal feeds and the food industry [24,25,26,27,28]. Soybean is cultivated extensively across diverse climates, and it plays a pivotal role in global nutrition and food security, contributing significantly to edible oil and plant-derived protein production [24,25]. Additionally, soybean enhances soil fertility through biological nitrogen fixation, making it an essential component of sustainable farming practices [29,30]. Soybean’s popularity stands for its wide adaptability and increasing cultivation across the globe. Worldwide production grew from around 160 million tons (Mt) on 70 M hectares (ha) in 1998 to 350 Mt on 125 M ha in 2018 [31]. The USA is one of the leaders in global soybean production, with significant contributions from other regions like China, Sub-Saharan Africa, Brazil, and Nigeria [32,33,34,35]. Soybean is also a major oil-producing crop in Bangladesh, occupying approximately 0.041 M ha with a yield of 0.064 Mt [36]. In Bangladesh, soybean cultivation has dramatically expanded from 5000 ha in 2005 [35] to 62,508.5 ha in 2018–2019 [36] due to its high market demand [1,27]. It is anticipated that soybean meal consumption in Bangladesh’s feed industry will reach 2.65 Mt in the year 2024–2025, marking a 0.15 Mt increase from the previous year (https://www.tbsnews.net/node/812956, accessed on 21 June 2024).

Despite its adaptability, soybean faces considerable challenges from climate variability, particularly in rainfed areas where unpredictable rainfall patterns threaten soybean growth and yield [37,38]. Water stress has a severe impact on morpho-physiological and yield traits in soybean including plant height, leaf area, pod yield, 100-seed weight, seed yield, and harvest index [38,39]. The extent of the effect depends on when the stress occurs and how intense the water shortage is. Drought stress alone is estimated to cause a 40% reduction in soybean yield [39]. The flowering stage and the period right after are the most crucial times for managing water stress in soybean [39,40,41]. Water requirements are twice as high during the reproductive phase compared to the vegetative phase in soybean [40,41]. Yield formation of soybean is affected by mild water deficits during the seedling phase, but it becomes even more sensitive to severe water shortages during the flowering and podding stages [42]. Therefore, efforts have been focused on developing drought-tolerant soybean varieties using genetic engineering, molecular breeding, and the mining of genetic factors at diverse growth stages [43]. However, selection of suitable genotypes with desirable traits that provide agronomic, breeding, and economic benefits continues to be a significant challenge [27,44]. Moreover, understanding the drought tolerance mechanisms of soybean is important for enhancing genetic advancements to develop climate-resilient soybean varieties [45,46]. This study aimed to evaluate the impact of drought stress on five previously unexplored soybean genotypes, along with two elite soybean varieties used as benchmarks. We assessed various morpho-physiological, biochemical traits, and yield attributes, to understand their response to water-limited conditions and ranked the genotypes based on their drought tolerance.

## 2. Results

### 2.1. Phenotypic Responses of Soybeans Genotypes Exposed to Drought Stress

In our study, morpho-physiological and biochemical traits revealed distinct responses among the seven investigated soybean genotypes [five promising advanced lines (G00001, G00046, G00135, BD2333, and PK472) and two popular Bangladeshi soybean varieties (BARI Soybean6 and BU Soybean2)] under control (100% FC) and drought (50% FC) conditions, as evidenced by statistically significant lower *p*-values compared to the *F*-value (Appendix A). Phenotypic variations revealed that drought stress (50% FC) led to substantial changes in the investigated soybeans’ genotypes including leaf yellowing and partial desiccation, in comparison to the control treatments (Figure 1A,B). The impacts of the investigated soybean genotypes under different water treatments on morphological and yield traits, including plant height (PH), leaf area plant^−1^ (LA), leaf number plant^−1^ (LP), branch number plant^−1^ (BP), pod number plant^−1^ (PP), seeds pod^−1^ (SP), 100-seed weight (HSW), and seed yield plant^−1^ (SY), are presented in Appendix A. The tallest plants were observed in genotype PK472 (63.10 cm) under control and genotype G00046 (45.60 cm) under drought treatment (Figure 1C). Genotype G00046 exhibited the highest PP under both control (31.80) and drought (20.80) conditions (Figure 1D). SY was the highest in G00046 (10.10 and 6.60 g), followed by G00001 (7.70 and 5.80 g), PK472 (7.60 and 3.30 g), BU Soybean2 (7.10 and 3.10 g), G00135 (4.70 and 1.50 g), BARI Soybean6 (3.50 and 1.40 g), and BD2333 (3.30 and 1.10 g) under control and drought conditions, respectively (Figure 1E and Appendix A). The heatmap displays the percent reduction in morphological traits across the investigated soybean genotypes under drought stress compared to control (Figure 1F). The minimum percent reductions were seen for the traits PH, PP, and SY in genotype G00001 (17.60, 12.20, and 24.60%) followed by G00046 (20.40, 34.60, and 34.60%), BU Soybean2 (22.90, 52.20, and 55.60%), PK472 (32.80, 46.70, and 56.80%), BARI Soybean6 (36.20, 72.70, and 60.70%), G00135 (36.60, 50.80, and 67.80%), and BD2333 (37.10, 85.90, and 67.00%, respectively) under drought conditions compared to the control conditions (Figure 1F and Appendix A). In contrast, genotype BD2333 showed the maximum percent reductions in PH (37.10%), PP (85.90%), SP (56.20%), and HSW (54.20%) under drought conditions, suggesting a greater sensitivity to drought stress compared to the other genotypes (Figure 1F and Appendix A).

### 2.2. Photosynthetic Responses of Soybean Genotypes Exposed to Drought Stress

Gas exchange features revealed that drought stress led to substantial changes in stomatal conductance (*gs*), photosynthetic rate (*Pn*), intercellular CO_2_ concentration (*C_i_*), transpiration rate (*E*), leaf temperature (LT), and instantaneous water use efficiency (WUEins) in comparison to the well-watered plants (Figure 2A–E and Appendix A). Genotype G00001 demonstrated maximum *gs* (0.45 and 0.15 mol H_2_O m^−2^ s^−1^), *Pn* (19.60 and 13.73 μmol CO_2_ m^−2^ s^−1^), and *E* (8.29 and 6.19 mmol H_2_O m^−2^ s^−1^) under both well-watered and drought conditions, indicating that G00001 managed water more efficiently under drought conditions compared to other genotypes (Figure 2A–C and Appendix A). On the other hand, when subjected to both control and drought conditions, genotype BD2333 (32.85 and 35.60 °C) recorded the highest LT followed by genotypes G00135 (31.38 and 34.62 °C) and PK472 (31.10 and 33.47 °C, respectively) (Figure 2D and Appendix A). In contrast, a better leaf thermal balance was noticed in G00001 as documented by a significantly lower level of LT (26.61 and 27.48 °C in control and drought conditions, respectively) (Figure 2D and Appendix A). The heatmap illustrates the percent reduction in leaf gas exchange traits under drought stress compared to control conditions across the investigated soybean genotypes (Figure 2E). The minimal percent decreases in *gs* (66.80%), *Pn* (30.00%), and *E* (25.30%) and the minimal percent increase in LT (3.30%) under drought conditions compared to the control conditions were observed in genotype G00001 (Figure 2E and Appendix A). In contrast, BD2333 exhibited the greatest percent reductions in *gs* (90.50%), *Pn* (71.00%), and *E* (59.40%), indicating greater susceptibility to water scarcity (Figure 2E and Appendix A).

### 2.3. Plant Water Relations of Soybean Genotypes Exposed to Drought Stress

To elucidate the genotypic role in plant adaptation under water stress, genotype–specific responses focused on relative water content (RWC), water saturation deficit (WSD), water retention capacity (WRC), water uptake capacity (WUC), and xylem exudation rate (XER) under different water regimes are presented in Figure 3A–D and Appendix A. Genotype BD2333 maintained a high RWC (91.21%) under control conditions, but it had the most substantial reduction (34.40%) under water–stressed condition, indicating its sensitivity to drought. Conversely, G00046 (10.50%) showed a tiny reduction in RWC under water stress, followed by G00001 (16.50%), BARI Soybean6 (24.20%), G00135 (25.40%), BU Soybean2 (28.40%), PK472 (29.10%), and BD2333 (34.40%) (Figure 3A, D and Appendix A). WSD values showed how well the genotype could handle water stress, but these values were not important for plants grown with adequate water. Under water-stressed conditions, BD2333 showed maximum WSD (40.16%), while the lowest WSD (22.37%) was recorded in genotype G00046 (Figure 3B and Appendix A). XER served as a critical indicator of responses to drought stress. The highest XER was found in genotype G00046 (0.50 g h^−1^) with subsequent ranks being BD2333 (0.45 g h^−1^), G00001 (0.41 g h^−1^), PK472 (0.39 g h^−1^), and BARI Soybean6 (0.38 g h^−1^) under control conditions. In addition, genotype G00001 (0.25 g h^−1^) exhibited maximum XER followed by G00046 (0.20 g h^−1^), PK472 (0.21 g h^−1^), BD2333 (0.19 g h^−1^), BU Soybean2 (0.19 g h^−1^), BARI Soybean6 (0.12 g h^−1^), and G00135 (0.10 g h^−1^) under drought conditions (Figure 3C and Appendix A). The heatmap illustrates the percent reduction in RWC, WSD, WRC, WUC, and XER across the investigated soybean genotypes under drought stress compared to control conditions (Figure 3D). Under drought stress compared to control conditions, the minimum percent reduction in RWC was observed in genotype G00046 (10.50%), followed by G00001 (16.50%), BARI Soybean6 (24.20%), G00135 (25.40%), BU Soybean2 (28.40%), and PK472 (29.10%). In contrast, BD2333 exhibited the maximum percent reduction in RWC (34.40%) and the highest percent increase in WSD (290.40%) (Figure 3D and Appendix A).

### 2.4. Levels of Photosynthetic Pigments, Proline, and Malondialdehyde in the Leaves of Soybean Genotypes Exposed to Drought Stress

To understand the physiological changes in the investigated genotypes under different water regimes, genotype-specific responses were examined, focusing on chlorophyll (Chl) *a*, Chl *b*, total Chls, carotenoids, proline, and malondialdehyde (MDA) (Figure 4A–G and Appendix A). Drought stress caused a reduction in Chl *a*, Chl *b*, total Chls, and carotenoids, and an increase in proline and MDA contents in all tested genotypes compared to the control plants (Figure 4A–F and Appendix A). Despite the drought stress, genotype G00001 had the highest levels of Chl *a*, total Chls, and carotenoids, with values of 3.05, 4.23, and 7.34 mg g^−1^ FW, respectively (Figure 4A–F and Appendix A), indicating a strong adaptation to water scarcity through enhanced photosynthetic efficiency. On the other hand, genotype BD2333 showed a statistically significant increase in Chl *a* (4.98 mg g^−1^ FW), Chl *b* (2.34 mg g^−1^ FW), and total Chls (7.32 mg g^−1^ FW) compared to the other investigated genotypes under control conditions (Figure 4A–D and Appendix A). Proline and MDA are used as stress biomarkers, with proline indicating an osmoprotectant, while MDA serves as a marker of oxidative stress and lipid peroxidation [15,16,17,18,19]. Among the investigated genotypes, BD2333 had the highest levels of proline (3.70 and 11.60 μg g^−1^ FW) and MDA (5.34 and 20.29 μmol g^−1^ FW, respectively) under both control and drought conditions, while genotype G00001 exhibited lower levels of proline (4.49 μg g^−1^ FW) and MDA (3.70 μmol g^−1^ FW) under drought conditions (Figure 4E,F and Appendix A). The heatmap depicts the percent reductions in Chl *a*, Chl *b*, total Chls, carotenoids, proline, and MDA under drought stress relative to control conditions across the investigated soybean genotypes (Figure 4G and Appendix A). Under drought stress, a substantial percent increase in proline was observed in BARI Soybean6 (532.30%), followed by BU Soybean2 (304.20%), PK472 (271.70%), BD2333 (213.20%), G00135 (184.40%), and G00046 (159.00%) (Figure 4G and Appendix A). Genotype G00001 exhibited the lowest accumulation of proline and MDA under drought conditions (Figure 4G and Appendix A), indicating a reduced level of oxidative stress response compared to the other genotypes. Conversely, genotype BD2333 exhibited the greatest percent reductions in Chl *a* (79.50%), total Chls (72.70%), and carotenoids (56.70%), along with the highest increase in MDA (280.30%) compared to the other genotypes, indicating a higher sensitivity to drought stress (Figure 4G and Appendix A).

### 2.5. Correlation between Studied Traits under Control and Drought Stress Conditions

The relationships between the studied traits under both control and drought stress conditions are shown in Figure 5 and Appendix A. Under the control conditions, the correlation analysis showed a strong positive relationship between LP and BP (*r* = 0.98 **), and a strong negative relationship was observed between *gs* and LT (*r* = −0.94 **). PH was negatively associated with LA (*r* = −0.78 *). SP had significant positive associations with WSD (*r* = 0.87 *) and WUC (*r* = 0.95 **), and a negative association with WRC (*r* = −0.87 **). HSW correlated positively with *Pn* (*r* = 0.80 *), *gs* (*r* = 0.88 **), and *E* (*r* = 0.88 **), and negatively with *Ci* (*r* = −0.84 *), LT (*r* = −0.84 *), WUEint (*r* = −0.76 *), and WUEins (*r* = −0.81 *). SY had a strong positive correlation with *Pn* (*r* = 0.85 *) and a negative one with RWC (*r* = −0.778 *). RWC positively correlated with WRC (*r* = 0.88 **) and negatively with WSD (*r* = −0.92 **) and WUC (*r* = −0.92 **). WSD was positively associated with WUC (*r* = 0.96 **) but negatively with WRC (*r* = −0.89 **) and LT (*r* = −0.76 *). WRC positively correlated with LT (*r* = 0.90 **) and negatively with WUC (*r* = −0.76 *). *Pn* showed positive correlations with *gs* (r = 0.92 **) and *E* (r = 0.97 **), and negative correlations with *Ci* (r = −0.91 **), LT (r = −0.91 **), WUEins (r = −0.79 *), and MDA (r = −0.78 *). The trait gs was positively correlated with *E* (r = 0.98 **) and negatively with *Ci* (r = −0.79 *), LT (*r* = −0.94 **), WUEint (*r* = −0.86 *), and WUEins (*r* = −0.87 *). *Ci* showed positive correlations with LT (*r* = 0.82 *), WUEins (*r* = 0.78 *), and carotenoids (*r* = 0.82 *), but negative ones with *E* (*r* = −0.89 **). *E* was only negatively correlated with LT (*r* = −0.94 **), WUEint (*r* = −0.78 *), and WUEins (*r* = −0.87 **). LT had positive correlations with WUEins (*r* = 0.76 *) and MDA (*r* = 0.87 *), while WUEint positively correlated with WUEins (*r* = 0.88 **). Chl *a* correlated positively with Chl *b* (*r* = 0.78 *) and total Chls (*r* = 0.97 **), while Chl *b* also showed a positive correlation with total Chls (*r* = 0.91 **). Carotenoid level was positively correlated with proline (*r* = 0.80 *) under the control conditions (Figure 5A and Appendix A). On the other hand, there were also significant positive correlations among the agronomic traits under the drought stress conditions. The trait PP was positively correlated with SP (*r* = 0.81 *), SY (*r* = 0.87 *), Chl *b* (*r* = 0.91 **), and carotenoids (*r* = 0.77 *), and negatively with MDA (*r* = −0.76 *). PH exhibited a negative correlation with WUEint (*r* = −0.76 *), while traits such as LA, LP, and BP showed no significant correlations with the other studied traits. SP also showed significant positive correlations with HSW (*r* = 0.79 *), SY (*r* = 0.78 *), *Pn* (*r* = 0.81 *), *gs* (*r* = 0.80 *), *E* (*r* = 0.77 *), and carotenoids (*r* = 0.83 *). HSW displayed strong positive correlations with total Chls (*r* = 0.95 **), carotenoids (*r* = 0.95 **), Chl *a* (*r* = 0.93 **), *Pn* (r = 0.93 **), *gs* (*r* = 0.90 **), *E* (*r* = 0.89 **), and SY (r = 0.85), and strong negative correlations with MDA (*r* = −0.95 **), proline (*r* = −0.94 **), LT (*r* = −0.92 **), and *Ci* (*r* = −0.80 *). SY correlated positively with *Pn* (r = 0.91 **), *gs* (*r* = 0.86 *), *E* (*r* = 0.90 **), Chl *b* (*r* = 0.87 *), total Chls (*r* = 0.85 *), and carotenoids (*r* = 0.91 **), and negatively with *Ci* (*r* = −0.76 *), LT (*r* = −0.86 *), proline (*r* = −0.89 **), and MDA (*r* = −0.82 *). RWC correlated positively only with total Chls (*r* = 0.77 *), and negatively with WUC (*r* = −0.98 **) and WSD (*r* = −1.00 **). WSD showed a significant positive correlation with WUC (*r* = 0.98 **) and a negative one with total Chls (*r* = −0.77 *). XER was negatively correlated with WUEins (*r* = −0.85 *). *Pn* showed positive correlations with *gs* (*r* = 0.96 **), *E* (*r* = 0.99 **), Chl *a* (*r* = 0.90 **), total Chls (*r* = 0.96 **), and carotenoids (*r* = 0.95 **), and negative correlations with *Ci* (*r* = −0.89 **), LT (*r* = −0.98 **), proline (*r* = −0.92 **), and MDA (*r* = −0.86 *). *gs* had positive correlations with *E* (*r* = 0.98 **), Chl *a* (*r* = 0.89 **), total Chls (*r* = 0.93 **), and carotenoids (*r* = 0.95 **), and negative correlations with *Ci* (*r* = −0.84 *), LT (*r* = −0.99 **), WUEint (*r* = −0.78 *), proline (*r* = −0.92 **), and MDA (*r* = −0.81 *). *Ci* positively correlated with LT (*r* = 0.83 *), WUEint (*r* = 0.81 *), proline (*r* = 0.88 **), and MDA (*r* = 0.83 *) and negatively with total Chls (*r* = −0.77 *) and carotenoids (*r* = −0.76 *). *Ci* also showed positive correlations with Chl *a* (*r* = 0.88 **), WUEint (*r* = 0.94 **), and carotenoids (*r* = 0.94 **), and negative correlations with LT (*r* = −0.98 **), proline (*r* = −0.89 **), and MDA (*r* = −0.79 *). LT was positively correlated with proline (r = 0.91 **), MDA (r = 0.83 *), and negatively with Chl *a* (*r* = −0.93 **), total Chls (*r* = −0.97 **), and carotenoids (*r* = −0.97 **), while WUEint had a positive correlation with proline (*r* = 0.82 *). Chl *a* was strongly correlated with total Chls (*r* = 0.98 **), carotenoids (*r* = 0.91 **), and negatively with proline (*r* = −0.80 *), and MDA (*r* = −0.80 *), whereas total Chls showed a positive correlation with Chl *a* (*r* = 0.97 **) and negative correlations with proline (*r* = −0.87 *) and MDA (*r* = −0.85 *). The carotenoid level was negatively correlated with proline (*r* = −0.93 **) and MDA (*r* = −0.88 **), while proline showed a strong positive correlation with MDA (*r* = 0.95 **) under drought stress conditions (Figure 5B and Appendix A).

### 2.6. Stress Tolerance Indices Utilizing Seed Yield

To identify the genotypes best suited for water shortage conditions, we assessed the drought tolerance levels among the seven investigated soybean genotypes using various tolerance indices related to yield performance (Table 1). Indices such as tolerance (TOL), mean productivity (MP), geometric mean productivity (GMP), and harmonic mean (HM) were used to evaluate the overall performance and tolerance level of genotypes under drought conditions. The stress susceptibility index (SSI) and stress tolerance index (STI) were focused on each genotype’s sensitivity and resilience to drought stress, respectively, with lower SSI and higher STI values being preferable for desirable traits. Yield index (YI) and yield stability index (YSI) were evaluated to measure yield performance and stability under water stress, with higher values indicating better performance. The relative stress index (RSI) assessed adaptation to stress relative to normal conditions, while the percent reduction (PR) in seed yield over control indicated the extent of yield reduction due to drought stress, with lower values indicating better drought tolerance. In our study, PK472 recorded the highest TOL (4.30), followed by BU soybean2 (3.90) and G00046 (3.50) (Table 1). Other indices like MP, GMP, HM, STI, and YI were found to be highest in the G00046 genotype (8.35, 8.16, 7.98, 1.69, and 2.12, respectively) followed by the G00001 genotype (6.75, 6.68, 6.62, 1.14 and 1.86, respectively) (Table 1). Conversely, the YSI and RSI were maximum in G00001 (0.75 and 1.52, respectively), followed by G00046 (0.65 and 1.32, respectively) (Table 1). Meanwhile, the lowest SSI and PR values were observed in genotype G00001 (0.49 and 24.68, respectively), followed by G00046 (0.69 and 34.65, respectively) (Table 1). Comparing these indices, G00001 followed by G00046 demonstrated high productivity and favorable tolerance indices, while BD2333 showed exceptional robustness but with a lower yield under water stress (Table 1).

### 2.7. Ranking of Soybean Genotypes Utilizing Stress Tolerance Indices

Ranking the investigated soybean genotypes using various stress tolerance indices, facilitates the identification of better-suited genotypes for cultivation under drought stress conditions (Table 2). Each index sorted genotypes from the best (1) to the worst (7) performance. G00001 ranked the highest in TOL (1), SSI (1), YSI (1), and RSI (1), showcasing consistent performance under water stress conditions, while G00046 led in MP (1), GMP (1), HM (1), STI (1), and YI (1) (Table 2). G00001 had the most favorable mean ranking (1.55) across all indices, indicating superior productivity and adaptability compared to the other genotypes under water shortage conditions (Table 2). In contrast, BD2333, with the highest mean ranking (6.55) across all indices, appears to be the least suitable for drought stress conditions (Table 2).

## 3. Discussion

To ensure sustainable soybean production in the face of climate change, it is essential to develop drought-tolerant soybean varieties for drought prone areas [18,37]. Drought-tolerant genotypes often exhibit diverse phenotypic traits, physiological responses, and yield attributes when exposed to different water regimes [47,48]. In this study, we employed five promising soybean genotypes along with two elite varieties to assess their drought tolerance potential. Specifically, genotypes G00001, G00046, G00135, BD2333, and PK 472 were known to show high-yielding characteristics; however, their drought tolerance potential was unexplored. We found that morpho-physiological parameters, stress metabolites, and yield traits of the investigated soybean genotypes were significantly influenced by different water regimes (Appendix A). Comparative evaluation of the performance of the studied genotypes and varieties revealed that genotype G00001 exhibited the lowest percent reductions in PH, PP, and SY under drought (50% FC) conditions compared to the control (100% FC) conditions, followed by genotypes G00046, BU Soybean2, PK472, BARI Soybean6, G00135, and BD2333 (Figure 1A–F and Appendix A). Poudel et al. [37] similarly noted that drought stress in soybeans resulted in reductions in PH, SY, HSW, and seed number plant^−1^. Leaf gas exchange features of the investigated genotypes showed significant variation under water stress conditions, with the G00001 genotype experiencing the minimum decrease, while BD2333 showed the greatest decrease (Figure 2A–E and Appendix A). Among the leaf gas exchange parameters, *gs* is used as an important indicator of various physiological processes including photosynthesis in plants [49]. Under drought conditions, a decrease in RWC induces stomatal closure, leading to decreased *gs* and, consequently, a reduction in *Pn* [1,50]. Genotype G00001 had the smallest decrease in *gs*, suggesting that the stomata of G00001 were highly functional in ensuring better *Pn* under water shortage conditions. Genotype G00001 exhibited the smallest reductions in *Pn* and *E* and the minimum increase in LT compared to the corresponding control, indicating that G00001 used water more efficiently than the other investigated genotypes under water scarcity conditions. Under drought stress, a reduction in *E* and the ability to effectively regulate LT [51,52,53,54], WUEint, and WUEins provide insights into plant water use efficiency [54]. In line with our findings, better regulation of *gs*, *Pn*, and *E* along with an increase in LT [55] have also been observed under drought stress in tolerant soybean genotypes [1]. Consequently, when plants partially close their stomata, it helps them maintain a balance between conserving water and allowing sufficient gas exchange. In turn, they achieve higher productivity even when water availability is limited [56,57,58]. Plant water relations are the best indicators of a plant’s hydrological status, reflecting the physiological impacts of water scarcity on cellular metabolism [59]. Our study revealed that water stress significantly exacerbated the water relation features of the investigated soybean genotypes (Figure 3 and Appendix A). Genotype G00046, followed by G00001, exhibited the smallest decline in leaf RWC, WRC, and WUC, highlighting their better water relations relative to the more susceptible genotypes under drought stress conditions [60,61]. Mishra et al. [62] also observed that sensitive soybean genotypes experienced a greater decline in RWC compared to drought-tolerant ones. Our findings showed a substantial increase in the WSD of soybean genotypes under drought stress, indicating that the investigated soybean genotypes were subjected to a greater degree of water deficit [63]. Drought-tolerant genotypes showed a superior ability to extract surface water under stress conditions, while XER was higher in well-watered conditions compared to drought stress [54]. Our findings indicate that XER decreased significantly under drought stress across all tested genotypes, suggesting that XER is a useful indicator for measuring drought severity. Consistent with our findings, Fatema et al. [1] and Filek et al. [64] also observed similar trends for water relations in soybean genotypes. Drought-exposed soybean genotypes exhibited a decline in the levels of Chl *a*, Chl *b*, total Chls, and carotenoids (Figure 4A–D, G, and Appendix A), resulting in a significant negative impact on photosynthetic pigments in the leaves. Genotype G00001 showed the lowest reduction in Chl *a* and carotenoids, suggesting that G00001 is more resistant to losing photosynthetic pigments necessary for efficient photosynthesis under drought conditions [1]. Plants accumulate various metabolites like proline and MDA, which are often used as stress indicators under drought conditions [20,65]. Our results indicated that both proline and MDA levels increased significantly in all the tested soybean genotypes under drought stress conditions (Figure 4E–G and Appendix A). This suggests that drought-stressed plants suffer significant challenges to maintain cellular integrity and function under adverse environmental conditions such as drought [1,49,66,67]. Furthermore, correlations among plant traits often indicate potential allometric relationships in biological processes [68]. Our correlation analysis (Figure 5 and Appendix A) showed that the strongest and most significant positive correlation was between LP and BP, indicating that an increase in the number of leaves is associated with an increase in the number of branches. The trait HSW was positively correlated with *Pn*, *gs*, and *E,* whereas it was negatively correlated with *Ci* and LT. This implies that an increment in HSW was mediated by a better photosynthetic rate, stomatal conductance, and transpiration rate where better transpiration reduces LT and greater *P_n_* decreases *Ci* by assimilating CO_2_ in the photosynthetic process. The trait SY was positively correlated only with *Pn*, illustrating an increment in seed yield because of a better *P_n_* in the soybean genotypes. The positive correlation of *Pn* with *gs* and *E* implies that greater photosynthesis in tolerant genotypes was maintained through conductive stomata and improved transpiration rate. The trait Chl *a* had a positive correlation with Chl *b* and total Chls, and Chl *b* had also a positive correlation with total Chls. The trait PP exhibited positive correlations with SP and SY, highlighting that the more the PP, the higher the seeds and SY. Our findings also align with the previous work of Fatema et al. [1], which found that stem dry matter, pods, and seeds plant^−1^ contributed most to seed yield. Similarly, Poudel et al. [37] suggested that genotypes with a higher SY under well-watered conditions might not maintain the same yield under drought due to weaker relationships between yield attributes across different treatments. Assessment of stress tolerance indices indicated that genotype G00001 exhibited high tolerance indices, while BD2333 showed remarkably lower resilience under water shortage conditions (Table 1). From ranking the seven investigated soybean genotypes based on stress tolerance, it was clear that G00001 consistently excelled in TOL, SSI, YSI, and RSI, demonstrating stable performance in drought conditions (Table 2). These results pinpoint that G00001 possesses better adaptability under drought conditions compared to the other studied genotypes. Ahsan et al. [66] and Poudel et al. [37] also utilized drought stress tolerance indices to classify soybean cultivars as either drought–tolerant or sensitive. Among the seven studied genotypes, our results demonstrated that genotypes G00001 and BD2333 were the most drought-tolerant and drought-sensitive, respectively. This provides important genetic resources for studying the drought responses of soybean and for breeding programs to enhance soybean tolerance level to drought.

## 4. Materials and Methods

### 4.1. Experimental Setup and Management

The experiment was laid out at the abiotic stress research site of the Agronomy Department, Bangabandhu Sheikh Mujibur Rahman Agricultural University (24°09′ north latitude and 90°26′ east longitude), Gazipur, Bangladesh, from November 2021 to March 2022. Weather data including air temperature, relative humidity, precipitation, and evaporation were obtained from an on-site weather station during the experiment and are shown in Appendix A. The experiment was set up in a completely randomized design with five replications. Five promising soybean genotypes (G00001, G00046, G00135, BD2333, and PK 472) were selected based on previous findings that demonstrated their high yield performance under normal growth conditions. Two popular Bangladeshi soybean varieties (BARI Soybean6 and BU Soybean2) were also included in this study. Plastic pots (14 L; diameter 25 cm × height 28 cm) were filled with air-dried field soil mixed with cow dung at a 3:1 ratio by weight. Pots were then placed under a transparent polyhouse to protect the plants from rainwater. Recommended doses of fertilizers for soybean plants were applied [69], with all fertilizers except half of the urea applied initially, and the other half applied 25 days after emergence. Ten healthy seeds were sown in each pot at 5 cm depth and covered with soil by hand. Light irrigation was applied immediately after sowing to ensure uniform germination of soybean seeds. At the 2nd trifoliate leaf stage [20 days after seed sowing (DAS)], 5 uniform soybean seedlings were kept in each pot. All pots were arranged into control and drought treatment groups. Genotype-wise control groups of soybean plants were maintained with adequate watering (100% FC) throughout the growing period. Drought-treated pots were maintained to keep 50% FC on a volume/weight basis from the 2nd trifoliate leaf stage to maturity. All intercultural operations were carried out following standard soybean management practices. To protect the soybean plants from pests and diseases, Decis (Bayer Crop Science Limited, India) was applied at a concentration of 1 mL L^−1^ of water at 35 and 55 DAS.

### 4.2. Assessment of Studied Traits

After 25 days of drought exposure, the 2nd trifoliate leaves of 45-day-old soybean plants were collected to measure various attributes related to soybean phenotypic and physio-biochemical traits. Relative water content (RWC, %) was calculated using the equation [fresh weight (FW) − dry weight (DW)]/[turgid weight (TW) − DW)] × 100, following the detailed techniques stated by Rahman et al. [24]. Water saturation deficit (WSD), water retention capacity (WRC), and water uptake capacity (WUC) were also calculated using equations 100-RWC, TW/DW, and TW − FW/DW, respectively [1]. The xylem exudation rate (XER) was measured 5 cm above the base of the control and stressed plants. First, dry cotton and aluminum foil were weighed. A slanting cut was made on the stem with a sharp knife. The weighed cotton was placed on the cut surface and covered with aluminum foil. Sap exudation was collected from the stem for 1 h. The final weight of the cotton and aluminum foil with sap was then taken. The exudation rate was calculated by subtracting the weight of dry cotton and aluminum foil from the weight of the sap-containing cotton and aluminum foil and was expressed as g on an hourly basis [54]. The leaf area plant^−1^ (LA) was measured using an LI-3100C area meter (LI-COR Environmental, Lincoln, NE, USA). The photosynthetic parameters, including stomatal conductance (*gs*), photosynthetic rate (*Pn*), transpiration rate (*E*), intrinsic water use efficiency (WUEint), and instantaneous water use efficiency (WUEins) were estimated using LI-6400/XT portable photosynthesis systems (LI-COR Environmental, Lincoln, NE, USA). Photosynthetic features were measured at midday (11:00–14:00 h) for maximum ambient sunlight.

### 4.3. Quantification of Photosynthetic Pigments, Proline, and Malondialdehyde

The chlorophyll (Chl *a*, *b*, and total Chls) and carotenoid levels were measured, following the detailed techniques described by Arnon [70] for chlorophylls and Lichtenthaler and Wellburn [71] for carotenoids. Fresh leaf samples were subjected to extraction with 80% acetone (*v*/*v*) followed by spectrophotometric quantification. The absorbance measurements were taken at wavelengths of 663, 645, and 470 nm using a UV-Vis spectrophotometer (Thermo Fisher Scientific Inc., Waltham, MA, USA). The following formulas were used to calculate the concentrations of Chl *a*, Chl *b*, total Chls, and carotenoids:Chl *a* = [(12.7 × A_663_ − 2.69 × A_645_) × V]/1000 × W
Chl *b* = [(22.9 × A_645_ − 4.68 × A_663_) × V]/1000 × W
Total Chls = [(20.21 × A_645_) + (8.02 × A_663_) × V]/1000 × W
Carotenoids= [(1000 × A_470_) − (2.27 × Chl *a*) − (81.4 × Chl *b*)]/227
where, A_663_, A_645_, and A_470_ represent the optical densities of the chlorophyll extract at wavelengths of 663 nm, 645 nm, and 470 nm, respectively; V is the final volume (mL) of 80% acetone containing extracted Chls; and W is the weight of the fresh leaf sample in g. The results were expressed as mg g^−1^ fresh weight (FW).

The proline level was measured following the method of Bates et al. [72]. To quantify the proline level, the absorbance of the upper toluene layer was recorded at 520 nm using a UV-VIS spectrophotometer (Thermo Fisher Scientific Inc., Waltham, MA, USA). The amount of proline was quantified using the standard curve and calculated based on the following equation:Proline (μg g^−1^ fresh weight) = (μg mL^−1^ proline × volume of toluene × volume of sulfosalicylic acid)/(0.1 × 115.5), 
where, 0.1 is the sample weight (g) and 115.5 is the molecular weight of proline.

The level of MDA was determined following the method described by Anik et al. [20]. To quantify the MDA level, the absorbance was recorded at 532 nm and 600 nm using a UV-VIS spectrophotometer (Thermo Fisher Scientific Inc., Waltham, MA, USA), with non-specific turbidity corrected by subtracting the absorbance at 600 nm. The MDA content was calculated using its molar extinction coefficient of 155 mM^−1^ cm^−1^ and expressed as μmol g^−1^ FW.

### 4.4. Measurement of Agronomic Traits and Stress Tolerance Indices

The harvesting of soybean genotypes was carried out when the pods were completely dried. Data on growth, yield, and yield-related agronomic traits were collected according to replication, with each replicate comprising ten plants for each genotype. The investigated agronomic traits, along with their acronyms, measurement units, and data collection procedures, are described in Appendix A. The stress tolerance and susceptibility indices were calculated for the investigated genotypes using the formulae presented in Appendix A [73,74,75,76,77,78,79].

### 4.5. Statistical Analysis

We conducted an analysis of variance (ANOVA) to evaluate differences in the studied traits among the groups. To identify which specific groups differed significantly, we followed up with Fisher’s least significant difference (LSD) test, using a 5% significance level to ensure statistical reliability. For the ANOVA and Fisher’s LSD tests, as well as for generating the heatmaps, the ‘doebioresearch’ and ‘pheatmap’ packages in R version 4.1.3 were employed, respectively (https://www.r-project.org/, accessed on 17 March 2024). Additionally, drought tolerance indices utilizing seed yield of soybean genotypes were measured using iPASTIC [80] (https://manzik.com/ipastic/, accessed on 17 March 2024). 

## 5. Conclusions

Our study highlights the importance of identifying and utilizing drought-tolerant soybean genotypes to enhance soybean productivity under water scarcity conditions. Through comprehensive assessment of phenotypic, physio-biochemical, and yield traits, the current study highlighted that drought stress has significant impacts on soybean growth, physio-biochemical functions, and yield. Among the seven studied genotypes, G00001 emerged as a promising drought-tolerant genotype because of having the highest LA, SP, and HSW under water shortage conditions. Additionally, G00001 showed the highest *gs*, *Pn*, and RWC, indicating its physiological ability to cope with drought stress. Moreover, it exhibited the highest levels of total Chls and carotenoids, which together helped sustain photosynthetic performance under water shortage conditions. Furthermore, G00001 accumulated low levels of stress markers such as proline and MDA, suggesting its greater resiliency towards drought-induced negative impacts. Based on stress tolerance indices, genotype G00001 was identified as the most drought-tolerant among the studied genotypes. In contrast, genotype BD2333 showed the highest percentage reductions in almost all morpho-physiological traits and the greatest increase in MDA, indicating a greater sensitivity to drought conditions. Among the investigated soybean genotypes, G00001 represents a valuable genetic resource, which could be utilized as breeding material to enhance drought tolerance in elite soybean varieties. However, further study is warranted to elucidate the genetic and molecular potential governing its superior drought tolerance traits.

## Figures and Tables

**Figure 1 plants-13-02765-f001:**
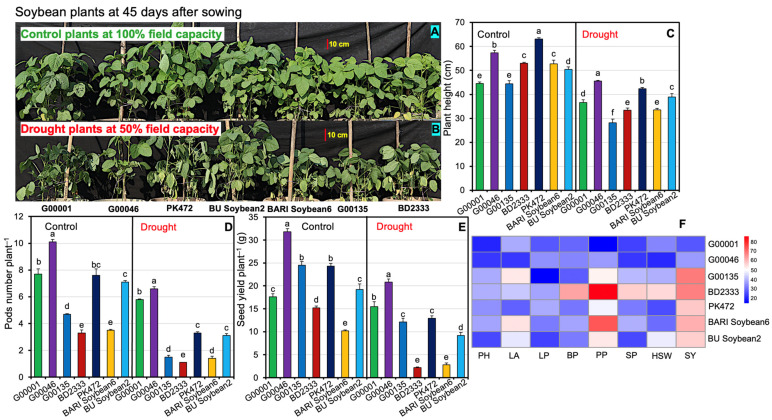
Effects of different water regimes on the morphological and yield traits of soybean genotypes. Plants were retained at 100% field capacity (FC) (control) (**A**) and at 50% FC (drought) (**B**) after 25 days of exposure to water-shortage treatment. Effect on plant height (**C**), pod number plant^−1^ (**D**), and seed yield plant^−1^ (**E**) of investigated soybean genotypes under different water regimes. (**F**) Heatmap illustrates the percent reductions in plant height (PH), leaf area plant^−1^ (LA), leaf number plant^−1^ (LP), branch number plant^−1^ (BP), pod number plant^−1^ (PP), seeds pod^−1^ (SP), 100-seed weight (HSW), and seed yield plant^−1^ (SY) in investigated soybean genotypes under drought compared to control conditions. Statistical analyses were conducted separately for control and drought treatments, with values derived from five biological replicates (*n* = 5; 10 plants per replicate). Bars symbolize means with standard errors for both control and drought-stressed soybean genotypes. Different letters are displayed on the bars to specify significant differences (*p* < 0.05) among the investigated soybean genotypes.

**Figure 2 plants-13-02765-f002:**
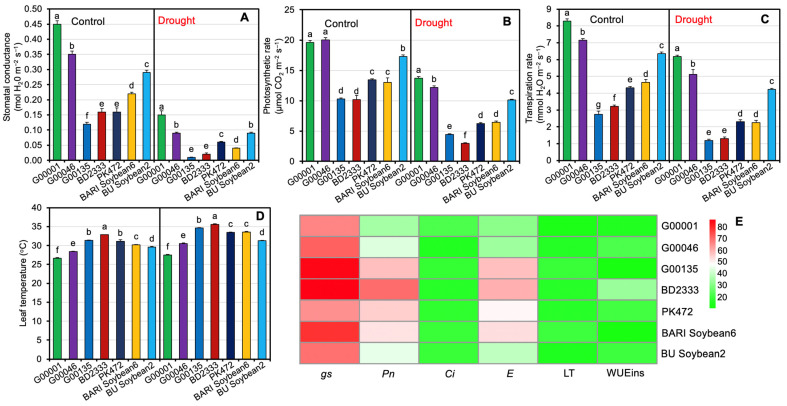
Photosynthetic responses of investigated soybean genotypes under different water regimes. Effect on stomatal conductance (**A**), photosynthetic rate (**B**), transpiration rate (**C**), and leaf temperature (**D**) of investigated soybean genotypes under different water regimes. (**E**) Heatmap illustrates the percent reductions in stomatal conductance (*gs*), photosynthetic rate (*Pn*), intercellular CO_2_ concentration (*Ci*), transpiration rate (*E*), leaf temperature (LT), and instantaneous water use efficiency (WUEins) in investigated soybean genotypes under drought compared to control conditions. Statistical analyses were conducted separately for control and drought treatments, with values derived from five biological replicates (*n* = 5; 10 plants per replicate). Bars symbolize means with standard errors for both control and drought-stressed soybean genotypes. Different letters are displayed on the bars to specify significant differences (*p* < 0.05) among the investigated soybean genotypes.

**Figure 3 plants-13-02765-f003:**
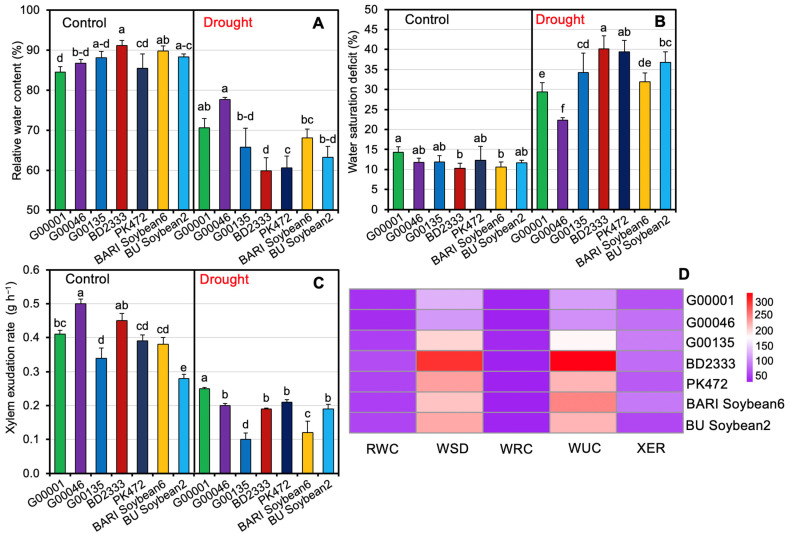
Effect on relative water content (**A**), water saturation deficit (**B**), and xylem exudation rate (**C**) of investigated soybean genotypes under different water regimes. (**D**) Heatmap illustrates the percent reductions in relative water content (RWC), water saturation deficit (WSD), water retention capacity (WRC), water uptake capacity (WUC), and xylem exudation rate (XER) of investigated soybean genotypes under drought compared to control conditions. Statistical analyses were conducted separately for control and drought treatments, with values derived from five biological replicates (*n* = 5; 10 plants per replicate). Bars symbolize means with standard errors for both control and drought-stressed soybean genotypes. Different letters are displayed on bars to specify significant differences (*p* < 0.05) among investigated soybean genotypes.

**Figure 4 plants-13-02765-f004:**
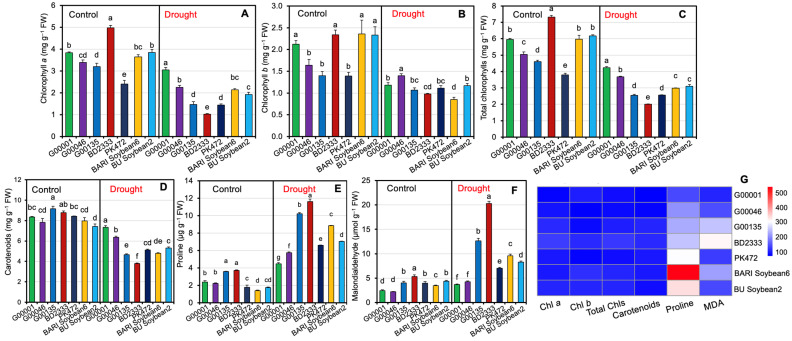
Changes in levels of photosynthetic pigments, proline, and malondialdehyde in leaves of investigated soybean genotypes under different water regimes. Effect on chlorophyll *a* (**A**), chlorophyll *b* (**B**), total chlorophylls (**C**), carotenoids (**D**), proline (**E**), and malondialdehyde (**F**) of investigated soybean genotypes under different water regimes. (**G**) Heatmap illustrates the percent reductions in chlorophyll *a* (Chl *a*), chlorophyll *b* (Chl *b*), total chlorophylls (Total Chls), carotenoids, proline, and malondialdehyde (MDA) of soybean genotypes under drought compared to control conditions. Statistical analyses were conducted separately for control and drought treatments, with values derived from five biological replicates (*n* = 5; 10 plants per replicate). Bars symbolize means with standard errors for both control and drought-stressed soybean genotypes. Different letters are displayed on the bars to specify significant differences (*p* < 0.05) among investigated soybean genotypes.

**Figure 5 plants-13-02765-f005:**
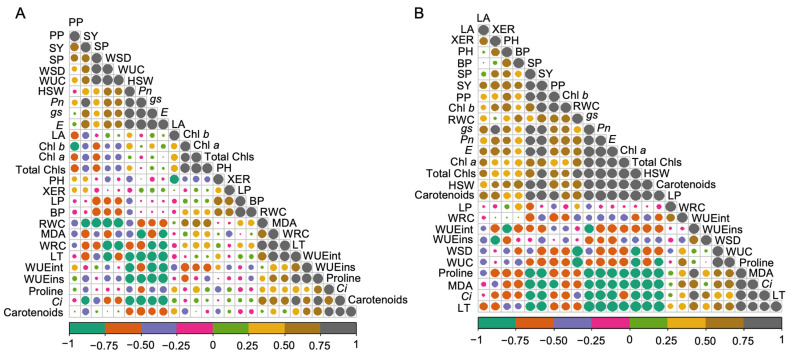
Pearson’s correlation coefficient among studied parameters under control (**A**) and drought stress (**B**) conditions. The correlations (positive to negative) among the traits were visualized by the color range green to gray in both control and drought stress conditions. Strong positive (+1) and negative (−1) associations between the two variables were presented in grey and green colored circles, respectively. Larger circles indicate a highly strong correlation, and smaller circles indicate a weaker relationship between the two traits. BP, branch number plant^−1^; Chl *a*, chlorophyll a; Chl *b*, chlorophyll b; *Ci*, intercellular CO_2_ concentration; *E*, transpiration rate; *gs*, stomatal conductance; HSW, 100-seed weight; LA, leaf area plant^−1^; LP, leaf number plant^−1^; LT, leaf temperature; MDA, malondialdehyde; PH, plant height; PP, pod number plant^−1^; *Pn*, photosynthetic rate; RWC, relative water content; SP, seeds pod^−1^; SY, seed yield plant^−1^; Total Chls, total chlorophylls; WRC, water retention capacity; WSD, water saturation deficit; WUC, water uptake capacity; WUEint, intrinsic water use efficiency; WUEins, instantaneous water use efficiency; XER, xylem exudation rate.

**Table 1 plants-13-02765-t001:** Stress tolerance (TOL), mean productivity (MP), geometric mean productivity (GMP), harmonic mean (HM), stress susceptibility index (SSI), stress tolerance index (STI), yield index (YI), yield stability index (YSI), relative stress index (RSI), and percent reduction (PR) indices of soybean genotypes.

Genotypes	TOL	MP	GMP	HM	SSI	STI	YI	YSI	RSI	PR
G00001	1.90 f	6.75 b	6.68 b	6.62 b	0.49 f	1.14 b	1.86 a	0.75 a	1.52 a	24.68 g
G00046	3.50 c	8.35 a	8.16 a	7.98 a	0.69 e	1.69 a	2.12 a	0.65 b	1.32 b	34.65 f
G00135	3.20 d	3.10 d	2.66 e	2.27 e	1.35 b	0.18 e	0.48 bc	0.32 d	0.64 f	68.09 b
BD2333	3.20 d	1.70 f	0.57 g	0.19 g	1.93 a	0.01 g	0.03 c	0.03 e	0.06 g	96.97 a
PK472	4.30 a	5.45 c	5.01 c	4.60 c	1.12 d	0.64 c	1.06 b	0.43 c	0.87 d	56.58 d
BARI soybean6	2.10 e	2.45 e	2.21 f	2.00 f	1.19 c	0.12 f	0.45 bc	0.40 c	0.81 e	60.00 c
BU soybean2	3.90 b	5.05 c	4.66 d	4.30 d	1.11 d	0.55 d	1.00 b	0.44 c	0.89 c	55.71 e
SD	0.88	2.41	2.65	2.73	0.46	0.61	0.77	0.23	0.47	23.36

Statistical analyses were conducted using values from three biological replicates. Distinct alphabetical symbols denote significant differences (*p* < 0.05) among genotypes for the traits analyzed, with a > b > c > d > e > f > g. SD, standard deviation.

**Table 2 plants-13-02765-t002:** Ranking of soybean genotypes using stress tolerance (TOL), mean productivity (MP), geometric mean productivity (GMP), harmonic mean (HM), stress susceptibility index (SSI), stress tolerance index (STI), yield index (YI), yield stability index (YSI), and relative stress index (RSI) indices.

Genotypes	TOL	MP	GMP	HM	SSI	STI	YI	YSI	RSI	Mean
G00001	1	2	2	2	1	2	2	1	1	1.55
G00046	5	1	1	1	2	1	1	2	2	1.77
G00135	4	5	5	5	6	5	5	6	6	5.22
BD2333	3	7	7	7	7	7	7	7	7	6.55
PK472	7	3	3	3	4	3	3	4	4	3.77
BARI soybean6	2	6	6	6	5	6	6	5	5	5.22
BU soybean2	6	4	4	4	3	4	4	3	3	3.88

## Data Availability

All data generated or analyzed during this study are included in this published article.

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
