# Peer review of "Differential Drought Responses of Soybean Genotypes in Relation to Photosynthesis and Growth-Yield Attributes"

_plants, 2024, doi:10.3390/plants13192765_

Round 1

Reviewer 1 Report

Comments and Suggestions for Authors

The author described the Differential drought responses of soybean genotypes: exploring phenotypic traits, photosynthesis, and yield attributes is a novel study, also drafted and arranged the data in an efficient way for readers but after going through I have some major concerns, and I hope the author can fix it during the revision stage. 

1. I suggest removing the (:) colon from the title, also too many commas (,) in one title is not scientifically sound. Fix this error.

2. The abstract is well described but I suggest if the author can add the important significant values then it will be easy for the readers to read just the abstract to understand the summary of your work.

3. Line 19-21, Please improve the text.

4. The introduction section is poorly written, I suggest improving the content and add more related literature, the current introduction text is not enough. 

5. The presentation of results is excellent however, I suggest adding the SD values and statistical letters in Tables 1 and 2.

6. The discussion section is well written but the supporting literature is not enough, so I suggest adding some recent literature.

7. Also, I suggest adding correlation analysis, which will enhance the effect of your treatments. 

Author Response

Please find an attachment

Reviewer 2 Report

Comments and Suggestions for Authors

The present study “Differential drought responses of soybean genotypes: exploring phenotypic traits, photosynthesis, and yield attributes” investigated response of seven promising soybean genotypes to identify their drought tolerance potential by exposing them to water shortage conditions from 2nd trifoliate leaf stage to maturity. The present work is written well and gives insights to identify the drought tolerance potential of soybean cultivars.

The abstract section starts with a strong rationale, however, it is written in very descriptive form. Revise the abstract section and add numerical values of study findings in this section.

Some abbreviations including HSW, gs, and Pn are not fully explained in the abstract. Please add the full names when first writing the abbreviations throughout the manuscript.

In the introduction add information about global soybean production and also add its nutritional value.

Line 86-90: The following lines should be deleted from the text.

The information about the field capacity levels must also be included in the abstract section of revised version.

Line 111: The least percent reductions, please revise this line add least percent reduction in revised paper. It will provide a better idea to reader how much reduction occurred.

The axis of the figures are not clear, please revise the figures and increase the size of the axis. The same comment for Figure 2 and other figures given in the manuscript.

The results are written in very descriptive form, please thoroughly revise the results section and re-write the results by adding numerical values and also try to reduce the volume of results.

In Table 1: Add letters to show the significance with ±SE or SD or use LSD values. The same comment could be applied to Table 2 as well.

I am not satisfied with the discussion section. There is a big tendency to repeat the results, please revise this section carefully add more logical reasoning and avoid using to many results in this section.

Comments on the Quality of English Language

Minor changes are needed. 

Author Response

Please find an attachment

Reviewer 3 Report

Comments and Suggestions for Authors

review plants-3171098

Comments to authors

Authors studied the morpho-physiological and biochemical responses of soybean lines and standard varieties to water deficit and determined their drought tolerance using yield-based drought stress indices.

-Line 115: Check the PH, SP reduction % for BD2333 variety in Figure 1. These values were not the highest. HWG can be a typos or HSW?

-Line 145: It is not sure LT decreased in G0001variety. I think this was slightly elevated compared to the control. This need to be checked.

-Lines 147-149: TableS3 is correct not Table S2. Correct it.

-Lines 184-185: WER is probably a typo, should XER be understood? If so, the finding is not true for BD2333 variety i.e. the rate of decline was not the largest. Check and correct it.

-Line 207:  How did you compare the improvement in chlorophyll content of BD2333 to other varieties or to what? 

Materials and methods section:

-371 -372 row: What is WSD, WRC, WUC, XER? The full name should be given here, not just the abbreviation of the property under consideration.

-Line 383: The full names of the abbreviations gs, Pn and E should also be given

-Line 383: WUEins and WUEins should be explained and why they were measured?

-Line 388: The methodological part of sub-chapter 4.2.1 should be described in a little more detail, it is not enough to refer only to the authors (Arnon, Lichtenthaler and Wellburn etc.).

Photosynthetic pigments, proline, malondialdehyde (MDA) were measured on a spectrophotometer at what wavelength and expressed as fresh weight or dry weight?

Author Response

Please find an attachment

Round 2

Reviewer 2 Report

Comments and Suggestions for Authors

The authors have substantially improved the MS as per my suggestions. Therefore, it can be sent for printing.

Thanks 

Comments on the Quality of English Language

Minor changes are needed.